# Prevalence and Impact of Vitamin D Deficiency in Critically Ill Cancer Patients Admitted to the Intensive Care Unit

**DOI:** 10.3390/nu13010022

**Published:** 2020-12-23

**Authors:** Nina Buchtele, Elisabeth Lobmeyr, Julia Cserna, Christian Zauner, Gottfried Heinz, Gürkan Sengölge, Wolfgang R. Sperr, Thomas Staudinger, Peter Schellongowski, Philipp Wohlfarth

**Affiliations:** 1Intensive Care Unit 13i2, Department of Medicine I, Medical University of Vienna, 1090 Vienna, Austria; nina.buchtele@meduniwien.ac.at (N.B.); elisabeth.lobmeyr@meduniwien.ac.at (E.L.); julia.cserna@hotmail.com (J.C.); wolfgang.r.sperr@meduniwien.ac.at (W.R.S.); thomas.staudinger@meduniwien.ac.at (T.S.); philipp.wohlfarth@meduniwien.ac.at (P.W.); 2Intensive Care Unit 13H1, Department of Medicine III, Division of Gastroenterology & Hepatology, Medical University of Vienna, 1090 Vienna, Austria; christian.zauner@meduniwien.ac.at; 3Intensive Care Unit 13H3, Department of Medicine II, Division of Cardiology, Medical University of Vienna, 1090 Vienna, Austria; gottfried.heinz@meduniwien.ac.at; 4Intensive Care Unit 13i3, Department of Medicine III, Division of Nephrology, Medical University of Vienna, 1090 Vienna, Austria; guerkan.sengoelge@meduniwien.ac.at; 5Department of Medicine I, Division of Hematology, Medical University of Vienna, 1090 Vienna, Austria; 6Stem Cell Transplantation Unit, Department of Medicine I, Medical University of Vienna, 1090 Vienna, Austria

**Keywords:** vitamin D, intensive care unit, cancer, oncology, hematology

## Abstract

Vitamin D deficiency is frequent in cancer patients and a risk factor for morbidity and mortality during critical illness. This single-center retrospective study analyzed 25-hydroxyvitamin D levels in critically ill cancer patients (n = 178; hematologic, n = 108; solid, n = 70) enrolled in a prospective ICU registry. The primary analysis was the prevalence of vitamin D deficiency (<20 ng/mL) and the severe deficiency (≤12 ng/mL). Secondary analyses included risk factors for vitamin D deficiency and its impact on ICU, hospital, and 1-year mortality. The prevalence of vitamin D deficiency and severe deficiency was 74% (95% CI: 67–80%) and 54% (95% CI: 47–61%). Younger age, relapsed/refractory disease, and a higher sepsis-related organ failure assessment (SOFA) score were independent risk factors for vitamin D deficiency (*p* < 0.05). After adjusting for relapsed/refractory disease, infection, the SOFA score, and the early need for life-supporting interventions, severe vitamin D deficiency was an independent predictor of hospital mortality (OR: 2.21, 95% CI: 1.03–4.72, *p* = 0.04) and 1-year mortality (OR: 3.40, 95% CI: 1.50–7.71, *p* < 0.01), but not of ICU mortality. Conclusion: Vitamin D deficiency is common in critically ill cancer patients requiring ICU admission, but its impact on short-term mortality in this group is uncertain. The observed association of severe vitamin D deficiency with the post-ICU outcome warrants clinical consideration and further study.

## 1. Introduction

Vitamin D is a fat-soluble steroid hormone and a key regulator of calcium and phosphate metabolism and bone health [1]. Signaling via its nuclear receptor influences multiple cellular pathways involving innate and adaptive immune responses, tissue proliferation, apoptosis, and differentiation [2]. Hypovitaminosis D, best reflected by low levels of 25-hydroxyvitamin D (25(OH) D), is a common phenomenon noted during periods of critical illness. This affects 40–70% of unselected patients treated on medical, surgical, or mixed intensive care units (ICUs) [3]. Several observational studies have indicated a relationship between 25(OH) D deficiency and unfavorable outcomes following ICU treatment, including prolonged ICU, stay, increased morbidity, and mortality [4,5,6,7]. While the mechanisms behind these associations are not fully understood, hypovitaminosis D appears to interact with other risk factors to promote organ dysfunction during various critical conditions, including sepsis [8], acute respiratory distress syndrome [9], and acute kidney injury [10].

An increasing number of patients worldwide live with active cancer due to aging populations, improved diagnostics, advances in supportive care, and more effective antineoplastic therapies [11]. These circumstances also impact critical care, as every sixth patient treated in European ICUs now carries a cancer diagnosis [12]. Furthermore, ICU admission rates may be as high as 20% during the treatment of some aggressive malignancies, such as acute leukemia [13]. The majority of cancer patients are referred to the ICU following elective surgery and can be managed routinely. Patients admitted for medical or surgical complications frequently suffer from extensive organ dysfunction and are at substantial risk for mortality [14]. Full-code ICU management is now advocated for many cancer patients with life-threatening complications as ICU survivors have the same long-term prognosis as patients without prior complications [13,15,16].

Low levels of 25(OH) D are common in cancer patients due to prolonged hospitalizations, avoidance of sunlight following radiation or cytotoxic therapy, and impaired nutritional status [17]. This finding may be even more pronounced in those patients requiring ICU treatment. In the absence of available data in the literature, this study aimed to analyze the prevalence of 25(OH) D deficiency (<20 ng/mL) [18] and severe deficiency (≤12 ng/mL) [19] in critically ill cancer patients requiring ICU admission. In addition, we aimed to explore risk factors for states of deficiency and assess the latter’s contribution to ICU, hospital, and 1-year mortality in this population.

## 2. Materials and Methods

This study included patients from an ongoing prospective cancer ICU registry (Institutional Review Board registration number: EK 1575/2013) operated at the Medical University of Vienna, Austria, who were enrolled between October 2014 and January 2019 (n = 341). The registry collects consecutive patients ≥18 years admitted to one of the four medical ICUs of our university hospital (1700 inpatient beds, 130 ICU beds, 48° latitude), who have a diagnosis of active cancer (i.e., diagnosis or treatment within the last five years) or who received an allogeneic hematopoietic stem cell transplantation at any time in the past. Only patients with an unscheduled ICU admission are eligible for enrollment. Patients consented to participate in the registry or were included with a waiver of consent in case of decease. The presented study was approved by the Institutional Review Board of the Medical University of Vienna (EK 1998/2017) and conducted under Good Clinical Practice guidelines and the amended Declaration of Helsinki. The primary analysis was the prevalence of 25(OH) D deficiency and severe deficiency in the study population. Secondary analyses included the identification of risk factors for 25(OH) D deficiency and severe deficiency and the association of both with ICU, hospital, and 1-year mortality.

### 2.1. Patients

For this study, we analyzed patients who had available results of 25(OH) D measurements within 72 h from ICU admission as part of the clinical routine. For the primary outcome analysis, 178 (52%) patients fulfilling the inclusion criterion were identified from the registry database. The characteristics of patients with and without available 25(OH) D measurements are shown in Appendix A. During the initial study period, no 25(OH) D supplementation protocols were in place at any of the participating ICUs. From early 2018, one of the participating ICUs enrolled patients with severe 25(OH) D deficiency in the VITDALIZE clinical trial (NCT03188796). Hence, secondary outcome analyses regarding mortality included only patients up to that time-point (n = 144).

### 2.2. Measurement of 25(OH) D Levels and Definitions

25(OH) D was measured from fresh blood samples on workdays using the Liaison^®^ 25(OH) vitamin D TOTAL Assay (DiaSorin, Saluggia, Italy). The technical specifications of this chemiluminescent immunoassay are available on the manufacturer’s website. Samples were obtained in the early morning hours in most patients during routine blood testing. We used cutoffs recommended by the Endocrine Society (Bethesda, Maryland, USA) to define 25(OH) D deficiency (<20 ng/mL) and severe deficiency (≤12 ng/mL) [20]. Similar cutoffs were used in the VIOLET [21] (only the former), the VITdAL-ICU [18], and the ongoing VITDALIZE [19] clinical trial, respectively. Patients were grouped by month of ICU admission to assess the impact of seasons on 25(OH) D levels (Spring: March, April, May; Summer: June, July, August; Autumn: September, October, November; Winter: December, January, February) [6].

### 2.3. Data Collection

Demographic and clinical data were collected prospectively by the study team and read into an electronic case report form after extraction from patient files. Baseline variables, including laboratory parameters shown in Table 1, were recorded at ICU admission. Reasons for ICU admission were documented according to the main symptoms present at ICU admission. The simplified acute physiology (SAPS) II [22] and the sepsis-related organ failure assessment (SOFA) score [23] were used to assess the severity of the acute illness and calculated using the worst manifestation of the respective parameters in the first 24 h after ICU admission. The Charlson comorbidity index (CCI) [24] was used to account for comorbidities. The diagnosis of a solid tumor, solid metastatic tumor, lymphoma, or leukemia, respectively, are already part of the CCI. The disease state of the underlying cancer was categorized according to the last documented response to the most recent therapy, with “naïve” denoting patients without any prior therapy. Furthermore, the goals of intensive care treatment (“full-code” versus “non-full-code”) as defined at ICU admission were recorded [25]. Documented life-supporting interventions included vasopressors, invasive mechanical ventilation, renal replacement therapy, and extracorporeal life support. The underlying registry collects data on ICU, hospital, and 1-year survival status as outcome variables, so this study used the same variables for secondary outcome analyses.

### 2.4. Presentation of Data and Statistics

Continuous variables are presented as median (interquartile range), dichotomous variables as absolute number (percentage). Confidence intervals (CI) of the primary analysis results were calculated using the Wilson score interval method. For univariate comparisons between groups, the Fisher’s exact test and the Mann–Whitney-U test were used as appropriate. The Wilcoxon signed-rank test was used to compare 25(OH) D levels before and after ICU admission. Optimal cutoff values were defined by Youden’s method. To analyze risk factors for 25(OH) D deficiency and severe deficiency and assess the contribution of both to ICU, hospital, and 1-year mortality, we conducted conditional multivariate logistic regression analyses incorporating variables associated with the dependent variable at *p* ≤ 0.2 in univariate analysis and deemed clinically relevant. Variables yielding *p* ≤ 0.1 were retained in the final models. The Hosmer–Lemeshow test was used to assess the goodness-of-fit of the logistic regression models. A two-sided *p*-value <0.05 was considered statistically significant. Results from secondary analyses were considered exploratory, so no adjustment for multiple testing was made. Analyses were done using SPSS Version 26.0 (IBM Corporation, Armonk, NY, USA).

## 3. Results

### 3.1. Patient Characteristics

The demographic and clinical characteristics of the included patients are shown in Table 1. Appendix A gives a detailed presentation of the cancer diagnoses in the study cohort. Acute myeloid leukemia (n = 37; 21%) and non-Hodgkin lymphoma (n = 36; 20%) accounted for the majority of cases in patients with hematologic malignancies (n = 108; 61%), whereas the lung (n = 13; 7%) and the head and neck region (n = 11; 6%) were the primary sites in patients with solid tumors (n = 70; 39%). The median time from cancer diagnosis to ICU admission was 15.4 (2.6–44) months, and most patients (n = 144; 81%) had received anticancer treatment 1 (0.3–16) months before study inclusion. Of the pretreated patients, 57 (40%) had received cytotoxic chemotherapy, 28 (19%) immunotherapies (including monoclonal antibodies) or small molecules, 27 (19%) surgery, 19 (13%) allogeneic hematopoietic stem cell transplantation, and 13 (9%) radiotherapy as the most recent line of treatment. Sixteen (9%) patients received their definite cancer diagnosis only while treated in the ICU. Patients were referred to the ICU either from the normal ward (n = 90; 51%), the emergency department (n = 44; 25%), the stem cell transplantation ward (n = 21; 12%), external ICUs (n = 20; 11%) or outpatient clinics (n = 3, 2%) after 3 (0–13) days following hospital admission. The median CCI was 3 (2–5) with congestive heart failure (n = 30; 17%), chronic pulmonary disease (n = 22; 12%), and moderate/severe renal disease (n = 16; 9%) as the most frequent comorbidities.

### 3.2. Intensive Care Unit Treatment, Hospital Survival, and 1-Year Survival

Acute respiratory failure (n = 121; 68%) was the primary reason for ICU admission, followed by infection (n = 107; 60%), shock (n = 87; 49%) and acute kidney injury (n = 73; 41%). In patients with infection, pneumonias (n = 65; 61%), primary bloodstream infections (n = 11; 10%), and gastrointestinal tract infections (n = 6; 6%) accounted for the most cases. Microbiological work-up identified a causative pathogen in 46% (n = 49) of documented infections. The focus of infection was unknown in 19 (18%) patients. The median SAPS II score at ICU admission was 52 (40–66); the median SOFA score was 10 (7–13). Most patients required vasopressors (n = 135; 76%) and invasive mechanical ventilation (n = 109; 61%). Renal replacement therapy and extracorporeal life support were used in 44 (25%) and 15 (8%) patients, respectively. Three out of four patients (n = 128; 72%) were eligible for full-code ICU management, meaning no a priori restrictions regarding the type or duration of any life-supporting intervention. During the ICU stay, 51 (29%) patients had documented neutropenia (<500/µL), and 26 (15%) received chemotherapy. Survival outcomes were analyzed in 144 (81%) patients. The median length of ICU stay was 11 (4–20) days. Ninety (63%) patients survived the ICU stay, and 71 (49%) could be discharged from the hospital alive. One year from ICU admission, the survival rate was 32% (n = 45; data available for 140 patients).

### 3.3. Prevalence and Risk Factors for Vitamin D Deficiency

The median 25(OH) D level measured within 72 h from ICU admission was 11.6 (7.2–21.9) ng/mL. The prevalence of 25(OH) D deficiency (<20 ng/mL) was 74% (95% CI: 67–80%); the prevalence of severe 25(OH) D deficiency (≤12 ng/mL) was 54% (95% CI: 47–61%). 25(OH) D levels obtained within 12 months before ICU admission were available in 39 (21%) patients. In these patients, 25(OH) D levels had decreased from 16.6 (9.8–27.4) ng/mL measured 2.3 (1.1–4.6) months before ICU admission to 11 (7.2–18) ng/mL thereafter (*p* < 0.01). Baseline variables associated with 25(OH) D deficiency in univariate analysis were younger age (59 (50–69) vs. 65 (55–73) years, *p* < 0.01; optimal cutoff: ≤61 years: 82% vs. 66%), relapsed or refractory disease (89% vs. 70%, *p* = 0.02), anticancer therapy during the last 12 months (80% vs. 64%, *p* = 0.02), and higher SOFA score (11 (8–14) vs. 9 (7–10), *p* < 0.01; optimal cutoff: >10:87% vs. 62%). There was a trend towards a higher rate of 25(OH) D deficiency in patients with infection (79% vs. 66%; *p* = 0.08) and with a longer duration of hospital stay before ICU admission (4 (0–16) vs. 2 (0–7) days; *p* = 0.08) (Appendix A). These variables were incorporated into a multivariate logistic regression model to predict 25(OH) D deficiency, but only age (OR: 0.95/year, 95% CI: 0.92–0.98, *p* < 0.01), relapsed or refractory disease (OR: 3.25, 95% CI: 1.04–10.19, *p* = 0.04), and the SOFA score (OR: 1.19/point, 95% CI: 1.07–1.33; *p* < 0.01) were retained in the final model (Table 2). The SOFA score was the only variable showing a borderline association with severe 25(OH D) deficiency (11 (8–13) vs. 9 (7–12), *p* = 0.08; optimal cutoff: >10:63% vs. 46%). The prevalence of 25(OH) D deficiency and severe deficiency were independent of the season of ICU admission (Table 1 and Appendix A).

### 3.4. Vitamin D Deficiency and Mortality

Contingency tables and unadjusted odds ratios for survival outcomes according to 25(OH) D levels are shown in Table 3. Neither 25(OH) D deficiency nor severe deficiency showed a statistically significant association with ICU mortality (42% vs. 26%, *p* = 0.12, and 44% vs. 30%, *p* = 0.12), but both predicted hospital mortality (57% vs. 34%, *p* = 0.02, and 59% vs. 41%, *p* = 0.04) and 1-year mortality (74% vs. 51%, *p* = 0.01, and 79% vs. 55%, *p* < 0.01) in univariate analysis (Appendix A). 25(OH) D levels ≤17 ng/mL and ≤12.4 ng/mL were the best discriminative values regarding hospital mortality (59% vs. 34%, *p* < 0.01) and 1-year mortality (79% vs. 54%, *p* < 0.01), respectively. Multivariate logistic regression analysis adjusting for relapsed or refractory disease, infection at ICU admission, the SOFA score, and the need for life-supporting interventions within 24 h from ICU admission was conducted to assess the independent contribution of 25(OH) D deficiency and severe deficiency to survival outcomes. Severe 25(OH) D deficiency was retained as an independent predictor for hospital mortality (OR: 2.21, 95% CI: 1.03–4.72, *p* = 0.04) and 1-year mortality (OR: 3.40, 95% CI: 1.50–7.71, *p* < 0.01) in the final models (Table 4). Furthermore, severe 25(OH) D deficiency (61% vs. 36%, *p* = 0.03) and disease state (*p* < 0.01) were the only variables showing a statistically significant association with 1-year mortality in ICU survivors (Appendix A).

## 4. Discussion

This retrospective observational study aimed to assess the prevalence of 25(OH) D deficiency and severe deficiency in critically ill cancer patients requiring ICU admission. We found respective rates of 74% and 54% in our cohort and identified younger age, relapsed or refractory disease, and a higher SOFA score as independent predictors of 25(OH) D deficiency. Finally, we showed that severe 25(OH) D deficiency (≤12 ng/mL) was an independent risk factor for hospital and 1-year mortality after adjusting for confounders in multivariate analysis. In ICU survivors, severe 25(OH) D deficiency and relapsed or refractory disease were the only variables associated with 1-year mortality.

Several observational studies have determined the prevalence of 25(OH) D deficiency in unselected patients treated in medical, surgical, or mixed ICUs to range between 40% and 70% [26]. Higher rates were found in burn patients [27], but other than them, risk groups for 25(OH) D deficiency in the ICU yet remain to be specified. Cancer patients seem to fit such a definition, as the 74% (95% CI: 67–80%) rate in our cohort is at the higher end of the mentioned range. Low levels of 25(OH) D are seen in many cancer patients already at the time of diagnosis [17]. Without adequate supplementation, deficiency usually aggravates further during anticancer treatment. For example, the rate of 25(OH) D deficiency in patients undergoing chemotherapy for breast cancer [28], colorectal cancer [29], or hematologic malignancies [30] may be as high as 90%. In line with this, most of our patients had received prior anticancer treatment, and we were able to identify relapsed or refractory disease as an independent risk factor for 25(OH) D deficiency in our cohort. Additionally, we found that a higher SOFA score and younger age were also independently associated with the primary outcome.

As measured by several scores, higher severity of the acute illness is a well-established predictor for low 25(OH) D levels at ICU admission [4,6,7]. The causality between these two factors is unclear. On one hand, 25(OH) D has convincingly been shown to reduce inflammation and progression of tissue damage in preclinical models of sepsis [31] and the acute respiratory distress syndrome [32]. On the other hand, fluid resuscitation, disrupted metabolism, reduced synthesis, and wasting of vitamin D and binding proteins can lead to rapidly falling 25(OH) D levels during critical illness [33,34]. The relationship between the presence of an active infection, as observed in 60% of our patients, and low 25(OH) D levels in the ICU is also well established, with the same limitation of unclear causal relations [3,31]. Our observation of lower 25(OH) D levels after ICU admission in patients with pre-admission levels obtained during the previous 12 months could have been attributable to direct interference of critical illness with vitamin D homeostasis, but also to other risk factors (see below) having acted in the meantime. Relatively low calcium and phosphate levels alongside high iPTH levels suggested the presence of true vitamin D deficiency in our cohort [35]. Still, iPTH levels were slightly (~10%) lower than expected from their known age-dependent relationship with 25(OH) D levels [36]. As one possible explanation for integrating these findings, we hypothesize that critical illness may have aggravated pre-existing vitamin D deficiency in several of our patients. In any case, the observed interaction between low levels of 25(OH) D, extensive organ dysfunction, and a high prevalence of infections and sepsis in critically ill cancer patients deserve attention, as even a modest benefit of a vitamin D-directed intervention may be clinically meaningful in this population [37].

Younger age also predicted low 25(OH) D levels in our cohort. Several studies of recent years have indicated higher rates of 25(OH) D deficiency in healthy young individuals and defied the classical notion of its predominance in the elderly [38,39,40]. The authors of these studies hypothesized that indoor occupation and lifestyle, sunscreen use, dietary habits, and less vitamin D intake might explain these findings. In the ICU, two large observational studies conducted by Venkatram et al. [4] and Braun et al. [5], respectively, discovered a similar association between younger age and 25(OH) D deficiency. Given that most of our patients had received prior treatment, we speculate that two cancer-specific additional factors might have contributed to our observation. First, younger patients might have been eligible for more aggressive antineoplastic therapies and thus have experienced more frequent and extended periods of hospitalizations. Second, patients with hematologic malignancies were slightly younger than solid cancer patients (59 vs. 64 years). Treatment of the former frequently incorporates the use of systemic corticosteroids, which is associated with 25(OH) D deficiency [41]. However, 25(OH) D levels did not differ according to the type of last anticancer therapy in our cohort (data not shown). Additional analyses will be needed to clarify these notions. In contrast to a study by Amrein et al. [6], the prevalence of 25(OH) D deficiency was independent of the season of ICU admission in our cohort. Hence, it appears that cancer and its sequelae exert a more significant effect on the 25(OH) D status than seasonal variation in UV light exposure.

Two randomized controlled trials (VIOLET [21] and VITdAL-ICU [18]) have assessed high-dose 25(OH) D supplementation in ICU patients with deficiency. While both trials showed no benefit for the intervention in patients with 25(OH) D levels <20 ng/mL, the VITdAL-ICU study observed a lower hospital mortality rate in a pre-defined subgroup analysis including patients with severe 25(OH) D deficiency (≤12 ng/mL). Due to these results, we decided to include the prevalence of severe 25(OH) D deficiency at the mentioned cutoff as a primary outcome variable and to explore its association with secondary outcomes. Indeed, we found that 25(OH) D levels ≤12 ng/mL predicted hospital and 1-year mortality after adjustment for other established risk factors present within 24 h following ICU admission. Other studies have defined different cutoffs for the deleterious effects of 25(OH) D deficiency on their respective outcome of interest (e.g., 15 ng/mL in cardiovascular disease [42]). The use of these different cutoffs might have led to divergent results in our cohort. Previous studies have shown that the severity of the acute illness and the need for life-supporting interventions are the major determinants of short-term outcomes in critically ill cancer patients [43]. In line with this, severe 25(OH) deficiency was not associated with ICU mortality. High-dose vitamin D supplementation in patients with severe 25(OH) D deficiency is currently being investigated in the ongoing VITDALIZE study, and its results are eagerly awaited [19]. Notably, some ICU patients have low levels of circulating 25(OH) and vitamin D binding protein, while levels of unbound 25(OH) D are not lower compared to controls [35]. For example, in a study by Palmer et al., concentrations of total 25(OH) were 35% lower in critically ill patients than in healthy controls, while free concentrations were not decreased [44]. We think that this factor must be kept in mind when interpreting the results of our study and others, including the mentioned randomized trials.

The observed lack of an association of 25(OH) D deficiency (<20 ng/mL) with survival outcomes in our study stands in contrast to several reports in the available literature [8,26,45]. However, most of the studies included unselected patients after scheduled and unscheduled ICU admissions who presented with less extensive organ dysfunction and a lower baseline mortality risk than our patients. Hence, we hypothesize that the generally high mortality in critically ill cancer patients may have diluted and limited a respective finding only to patients with severe deficiency. In this context, it is even more striking that severe 25(OH) D deficiency in our patients showed the strongest association with 1-year mortality. This finding may be explained by the overall impact of 25(OH) D deficiency on cancer-related mortality reported for several entities [46]. While further studies are needed, we think that the rate and the association of severe 25(OH) D deficiency with the post-ICU outcome observed in our cohort has clinical implications. Irrespective of the possible benefit of any future vitamin D-directed intervention in the ICU, we propose that the 25(OH) D status be assessed in critically ill cancer patients latest after discharge from the ICU. If 25(OH) D deficiency or severe deficiency is detected in the ICU, this information should reliably be handed over to the treating oncologist and hematologist, and 25(OH) D supplementation should be considered. This approach is supported by two meta-analyses of randomized controlled trials that revealed a 12% relative reduction in cancer-related mortality by vitamin D supplementation [47,48]. Of note, guidelines addressing 25(OH) D supplementation in cancer patients only target bone health [49], and two recent interventional protocols specifically aimed at improving response to treatment and cancer-related mortality have produced inconclusive results [50].

Our study has several strengths and limitations. It is the first report on 25(OH) D deficiency in critically ill cancer patients and, to our best knowledge, the largest of its kind involving a defined subgroup of ICU patients. Notably, 25(OH) D levels and the remaining data were obtained from a prospective cancer ICU registry and considered high-quality, although the present study is a retrospective study. 25(OH) D levels were available in only half of the registry patients, but we found no overt differences between patients with and without available 25(OH) D measurements, which rendered a relevant selection bias unlikely. Patients were enrolled in four different ICUs of a single-center, which may hamper the generalizability of our data. In addition, the underlying registry does not collect data on race categories or ethnicity. People of non-white origin have very likely been underrepresented, given our country’s demographics. The retrospective collection of 25(OH) D measurements precluded uniform sample collection and processing. However, levels were measured mostly during morning blood draws in the clinical routine, and all were collected within 72 h after ICU admission. Unfortunately, the sample size was too small to analyze our data for associations of individual cancer diagnoses or comorbidities with 25(OH) D deficiency. Lastly, many established risk factors (e.g., age, gender, race, smoking status, medications, reasons for ICU admission, renal replacement therapy, infections) affect 25(OH) D levels in critically ill patients and the general population [51]. Due to the lack of a control group, we could not assess the contribution of these factors to our observed primary outcome and, therefore, not evaluate an independent effect of cancer on 25(OH) D levels. Further studies to that respect are warranted.

## 5. Conclusions

In critically ill cancer patients admitted to the ICU, the rates of 25(OH) D deficiency and severe deficiency were 74% and 54%, respectively. A high level of awareness for 25(OH) D deficiency irrespective of younger age or season is warranted in these patients. Additionally, patients with more extensive organ dysfunction and relapsed or refractory disease must be considered at exceptionally high risk for the condition. Severe 25(OH) D deficiency was found as an independent contributor to hospital and 1-year mortality. Hence, 25(OH) D status should be assessed in ICU survivors, as they may benefit from 25(OH) D supplementation.

## Figures and Tables

**Table 1 nutrients-13-00022-t001:** Cohort characteristics.

	All Patients (n = 178)	25(OH) D ≥20 ng/mL (n = 47)	25(OH) D 12–20 ng/mL (n = 35)	25(OH) D ≤12 ng/mL (n = 96)
Sex, male	117 (66)	28 (60)	25 (71)	64 (67)
Age, years	61 (53–71)	65 (55–73)	56 (36–68)	60 (54–69)
Body mass index (kg/m^2^)	25.2 (21.9–29.1)	24.5 (20.8–28.5)	26.3 (23.9–28.7)	24.8 (22–30)
Season of ICU admission				
Spring	53 (30)	14 (30)	8 (23)	31 (32)
Summer	45 (25)	9 (19)	10 (29)	26 (27)
Autumn	42 (24)	14 (30)	10 (29)	18 (19)
Winter	38 (21)	10 (21)	7 (20)	21 (22)
Type of cancer				
Hematologic	108 (61)	26 (55)	26 (74)	56 (58)
Solid cancer	70 (39)	21 (45)	9 (26)	40 (42)
Disease state				
Naïve	34 (19)	11 (23)	3 (9)	20 (21)
Complete or partial remission	67 (38)	16 (34)	15 (43)	36 (37)
Relapsed or refractory	37 (21)	4 (9)	12 (34)	21 (22)
Undetermined/Unknown	40 (22)	16 (34)	5 (14)	19 (20)
Time from diagnosis, months	15.4 (2.6–44)	16 (1.9–54.4)	13.8 (2.9–37.9)	16.7 (2.6–38.4)
Time since last therapy, months	1 (0.3–16)	1.6 (0.3–45.2)	1.1 (0.4–15)	0.9 (0.2–6.3)
Therapy during last 12 months	106 (60)	21 (45)	24 (69)	61 (64)
Allogeneic HSCT	25 (14)	5 (11)	5 (14)	15 (16)
Charlson comorbidity index (CCI)	3 (2–5)	3 (2–5)	3 (2–4)	3 (2–5)
Hospital to ICU admission, days	3 (0–13)	2 (0–7)	4 (0–23)	3 (0–15)
Reasons for ICU admission				
Acute respiratory failure	121 (68)	34 (72)	24 (69)	63 (66)
Infection	107 (60)	23 (49)	25 (69)	60 (63)
Shock	87 (49)	21 (45)	19 (54)	47 (49)
Acute kidney injury	73 (41)	15 (32)	17 (49)	41 (43)
Neurological dysfunction	22 (12)	4 (9)	4 (11)	14 (15)
SAPSII at ICU admission	52 (40–66)	48 (43–58)	49 (34–71)	55 (40–67)
SOFA score at ICU admission	10 (7–13)	9 (7–10)	11 (8–15)	11 (8–13)
Life-supporting interventions				
Vasopressors	135 (76)	35 (75)	27 (77)	73 (76)
Invasive ventilation	109 (61)	25 (53)	26 (74)	58 (60)
Renal replacement therapy	44 (25)	7 (15)	9 (26)	28 (29)
Extracorporeal life support	15 (8)	3 (6)	3 (9)	9 (9)
25(OH)-D_3_ (ng/mL)	11.6 (7.2–21.9)	28.4 (24–31.8)	15.8 (13.4–19.4)	7.8 (5.4–9.9)
25(OH)-D_3_ (<20 ng/mL)	131 (74)			
25(OH)-D_3_ (≤12 ng/mL)	96 (54)			
Total calcium (mmol/L)	2.01 (1.87–2.10)	2.02 (1.91–2.12)	2.04 (1.88–2.13)	1.95 (1.86–2.08)
Ionized calcium (mmol/L)	1.1 (1.04–1.65)	1.09 (1.05–1.14)	1.13 (1.05–1.24)	1.1 (1.02–1.17)
Phosphate (mmol/L)	1.24 (0.89–1.56)	1.23 (0.9–1.42)	1.24 (0.89–1.50)	1.24 (0.87–1.71)
iPTH (pg/mL)	50.6 (30.3–115.1)	52.7 (32.4–113.4)	45.5 (25.3–79.3)	55.5 (30.8–140.3)
WBC (G/L)	12.2 (3–18)	12.8 (6.6–19.2)	10.2 (1.9–18.5)	12.2 (2.8–17.1)
Albumin (mg/dL)	25.7 (23.1–29.8)	25 (22.1–28.2)	27.2 (23.3–31.1)	25.7 (22.8–30.2)
Creatinine (mg/dL)	1.19 (0.8–2.2)	1 (0.8–1.67)	1.23 (0.81–2.43)	1.37 (0.75–2.48)
CRP (mg/dL)	20.4 (7–35.2)	18.4 (6.2–40.8)	23.3 (4.8–40)	21.3 (7.6–33.4)

Data are presented as absolute numbers (%) or median (interquartile range). Abbreviations: CRP = C-reactive protein; G/L = Giga per liters; HSCT = hematopoietic stem cell transplantation; ICU = intensive care unit; iPTH = intact parathyroid hormone; SAPS II = simplified acute physiologic score II; SOFA = sepsis-related organ failure assessment score; WBC = white blood cell count; inter-group comparisons between patients with and without 25(OH) D deficiency and severe deficiency, respectively, are shown in Appendix A.

**Table 2 nutrients-13-00022-t002:** Logistic regression analysis regarding 25(OH) D deficiency (<20 ng/mL).

**Variable (Univariate Analysis)**	**Odds Ratio**	**95% Confidence Interval**	***p***
Age	0.96/year	0.93 to 0.99	<0.01
Relapsed or refractory disease	3.62	1.21 to 10.85	0.02
Therapy during the last 12 months	2.29	1.16 to 4.51	0.02
Days from hospital to ICU admission	1.03/day	1.00 to 1.07	0.05
Infection	1.87	0.95 to 3.66	0.07
SOFA Score	1.16	1.05 to 1.27	<0.01
**Variable (Multivariate Analysis)**	**Odds Ratio**	**95% Confidence Interval**	
Age	0.95/year	0.92 to 0.98	<0.01
Relapsed or refractory disease	3.25	1.04 to 10.19	0.04
SOFA score	1.19/point	1.07 to 1.33	<0.01

Variables from the univariate analysis were included in the initial multivariate logistic regression model and eliminated backward at *p* > 0.1. Hosmer–Lemeshow goodness-of-fit test: χ^2^ = 4.91, df = 8, *p* = 0.77. Abbreviations: SOFA = sepsis-related organ failure assessment score.

**Table 3 nutrients-13-00022-t003:** Contingency tables and unadjusted odds ratios for survival outcomes according to 25(OH) D levels.

**25(OH) D**	**ICU Survival**		***p***	**Odds Ratio (Mortality)**	**95% Confidence Interval**
	**No**	**Yes**			
≥20 ng/mL	10 (26)	28 (74)		Reference	
12–20 ng/mL	10 (36)	18 (64)	0.43	1.56	0.54 to 4.48
≤12 ng/mL	34 (44)	44 (56)	0.10	2.16	0.93 to 5.06
**25(OH) D**	**Hospital Survival**		***p***	**Odds Ratio (Mortality)**	**95% Confidence Interval**
≥20 ng/mL	13 (34)	25 (66)		Reference	
12–20 ng/mL	14 (50)	14 (50)	0.22	1.92	0.71 to 5.22
≤12 ng/mL	46 (59)	32 (41)	0.02	2.85	1.27 to 6.42
**25(OH) D**	**1-Year Survival**		***p***	**Odds Ratio (Mortality)**	**95% Confidence Interval**
≥20 ng/mL	19 (51)	18 (49)			
12–20 ng/mL	17 (61)	11 (39)	0.61	1.46	0.54 to 3.96
≤12 ng/mL	59 (79)	16 (21)	<0.01	3.49	1.49 to 8.17

Data in the contingency tables are presented as absolute patient numbers and row-wise percentage. *P*-values are from Fisher’s exact test. ICU survival and hospital survival data were available for 144 patients, 1-year survival data were available for 140 patients, respectively.

**Table 4 nutrients-13-00022-t004:** Multivariate logistic regression analysis regarding survival outcomes.

Variables	Adjusted Odds Ratio	95% Confidence Interval	*p*
ICU Mortality			
Infection at ICU admission	2.81	1.30 to 6.08	<0.01
SOFA score	1.14/point	1.02 to 1.26	0.02
RRT within 24 h of ICU admission	3.56	1.12 to 11.34	0.03
Hospital Mortality			
Relapsed/refractory disease	2.99	1.14 to 7.87	0.03
Infection at ICU admission	2.48	1.15 to 5.36	0.02
IMV within 24 h of ICU admission	3.10	1.43 to 6.71	<0.01
RRT within 24 h of ICU admission	9.32	1.84 to 47.21	<0.01
Severe 25(OH) D deficiency (≤12 ng/mL)	2.21	1.03 to 4.72	0.04
1-Year Mortality			
Relapsed/refractory disease	4.44	1.91 to 16.56	0.03
Infection at ICU admission	3.16	1.41 to 7.07	<0.01
IMV within 24 h of ICU admission	2.54	1.11 to 5.82	0.03
Severe 25(OH) D deficiency (≤12 ng/mL)	3.40	1.50 to 7.71	<0.01

Results from univariate analyses regarding survival outcomes are shown in Appendix A. Variables included in the initial models: SOFA score, relapsed or refractory disease, infection at ICU admission, vasopressors within 24 h of ICU admission, invasive mechanical ventilation within 24 h of ICU admission, renal replacement therapy within 24 h of ICU admission, 25(OH) D deficiency (<20 ng/mL), severe 25(OH) D deficiency (≤12 ng/mL); backward elimination at *p* > 0.1. Hosmer–Lemeshow goodness-of-fit test: χ^2^ = 9.69, df = 8 *p* = 0.29 (ICU mortality); χ^2^ = 9.92, df = 8, *p* = 0.27 (hospital mortality); χ^2^ = 6.09, df = 8, *p* = 0.64 (1-year mortality). Abbreviations: ICU = intensive care unit; IMV = invasive mechanical ventilation; RRT = renal replacement therapy; SOFA = sepsis-related organ failure assessment score.

## Data Availability

The data presented in this study are available on request from the corresponding author. The data are not publicly available due to the regulations of the sponsor of this study (Medical University of Vienna).

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
