# Peer review of "Prevalence and Impact of Vitamin D Deficiency in Critically Ill Cancer Patients Admitted to the Intensive Care Unit"

_nutrients, 2020, doi:10.3390/nu13010022_

Round 1

Reviewer 1 Report

Review- nutrients- 1002024

In the single-center retrospective study, N Buchtele et al analyzed 25OHD levels in critically ill cancer patients enrolled in prospective ICU registry. They found that 74% of patient had 25OHD <20ng/mL) and 54% of patient had 25OHD ≤12ng/mL. After adjusting for relapsed/refractory disease, infection, the SOFA score, and the early need for life-supporting interventions, 25OHD ≤12ng/mL was an independent predictor of hospital mortality and 1-year mortality, but not of ICU mortality. The authors concluded that Vitamin D deficiency is common in ICU critically ill cancer patients.

General comments: There is a highly heterogenous patient population with so many factors would affect vitamin D binding protein (DPB) and 25OHD levels (1): age, gender, race, smoker, medications, cancer types (hemolytic vs solid), days in hospitals, types and duration of cancer treatments, reasons for ICU admission, renal replacement therapy, infection (sepsis)…, a control group would make it a much stronger study.

Specific comments:

  1. It looks like that days in hospital before ICU admission effect 25OHD levels (table1, P=0.08). Do you have 25OHD levels before admission to hospital (non-critical ill cancer patients) or in hospital before admitted (acute ill cancer patients) to ICU? Or any 25OHD levels difference within 24, 48 or 72 hours from ICU admission? Zaidan suggest the low 25OHD might reflect severity of critic ill (2) and in your case reflecting by SOFA. Thus, the 25OHD level before hospital or before ICU admission might reflect vitamin D nutrition status better.
  2. It is well known that there is high prevalence of vitamin D deficiency or low total 25OHD levels in critical ill ICU patients including infection and sepsis. In your study, infections effect 25OHD levels (p=0.08) in cancer ICU patient as other critical ill ICU patients. The author should explain why their finding is unique, may be add 25OHD ≤12ng/mL as a separated column in table 1 help your arguments. 
  3. Critically ill patients have low circulating 25(OH)D, and DBP and the free 25OHD remained relative unchanged (3,4). Palmer et al found that total 25(OH)D concentrations of critically ill patients were about 35% lower than the levels of controls but unbound 25(OH)D were not lower in these patients (4). The authors should cite the related literature in their introduction or discussion. It might be why that 2 randomized controlled trials (VIOLET and VITdAL-ICU) failed to show benefits of high-dose 25(OH) D supplementation in ICU patients with deficiency since low total 25OHD might ≠ vitamin D deficiency.
  4. Most cancer patients (n=144; 81%) had received anti-cancer treatment 1 (0.3-16) months before study. In table 1, therapy during last 12 months effects 25OHD levels (P=0.02); did any special anti-cancer treatment effect DBP or vitamin D metabolism as HIV therapy (1)?
  5. Young patient likely has hematologic malignancy and received special chemotherapy including steroid. Steroid treatment likely will decrease DBP levels and 25OHD levels (1,2) and the fact may explain young ICU cancer patient have lower total 25OHD levels.
  6. Do you have calcium, phos and iPTH levels ? Any black cancer patients?
  7. References:
    1. P Youselzadeh et al . Int J Endocrinol, vol. 2014; 2014. doi:10.1155/2014/981581

    2.J Zaidan, High-Dose Vitamin D3 for Critically Ill Vitamin D–Deficient Patients. N Engl J Med 2020; 382:1669-1671

    3.N Jassil, et al. Endocr Pract. 2017;23: 605-61

    4.D Palmer, et al, Unbound Vitamin D Concentrations Are Not Decreased in Critically Ill Patients. Intern Med J. 2020 Oct 11. doi: 10.1111/imj.15096. Online ahead of print.

Reviewer 2 Report

Abstract

  1. What does SOFA stand for?

Introduction

  1. Line 39 Moreover, signaling via its nuclear receptor influences multiple cellular pathways involved in innate and adaptive immune responses, tissue proliferation, apoptosis, and differentiation.

Please change into “Moreover, signaling via its nuclear receptor influences multiple cellular pathways involvinginnate and adaptive immune responses, tissue proliferation, apoptosis, and differentiation.”

  1. Moderate English language and style change are required.

  1. Line 64 In the absence of available data in the literature, this study thus aimed to analyze the prevalence of D deficiency (<20 ng/ml) and severe deficiency (≤12 ng/mL)…

Please insert reference.

Result

  1. Table 2 provides few information and thus is suggested as a supplement table.

  1. 3. Prevalence and risk factors for Vitamin D deficiency

Please provide data of univariate logistic regression analysis in Table 3.

  1. Table 4 Contingency tables for survival outcomes according to 25(OH) D levels.

Please try the tertiary method

25(OH)D

ICU survival

P

Odds ratio

No

Yes

≥20 ng/mL

10

38

Reference

20-12 ng/mL

≤12 ng/mL

34

44

0.009

2.93 (1.28, 6.72)

Please try the cut-off 25(OH)D levels ≥15 ng/ml.

(Vitamin D Deficiency: An Important, Common, and Easily Treatable Cardiovascular Risk Factor? 2008)

25(OH)D

ICU survival

P

Odds ratio

No

Yes

≥15ng/mL

Reference

<15ng/mL

  1. Line 258 We were surprised to find that younger age also predicted 25(OH) D deficiency in our cohort, as, in the general population, older people are usually more affected[1].

More factors might have contributed to the observation. Please see the following references.

Some studies demonstrate that younger age independently predicts hypovitaminosis D. (Age, Gender and Season Are Good Predictors of Vitamin D Status Independent of Body Mass Index in Office Workers in a Subtropical Region, Nutrients, 2020; Vitamin D insufficiency in Korea—A greater threat to younger generation: The Korea National Health And Nutrition Examination Survey (KNHANES) 2008. J. Clin. Endocrinol. Metab. 2011)

  1. Line 305 Although 25(OH) D levels were collected “retrospectively”,…

This point is the strength not a limitation. Please change into “Notably, 25(OH) D levels were obtained from a prospective cancer ICU registry and considered high quality although the present study is a retrospective study.

Round 2

Reviewer 1 Report

The authors addressed most of my comments and the revised manuscript has improved significantly. I have only a few minor comments.

  1. Calcium, ionized calcium, phos and iPTH levels in table 1. are welcome, but do these levels support vitamin D deficiency? The relative low calcium and phos levels with relatively high iPTH levels do suggest true vitamin D deficiency (Jessil et al).
  2. 25(OH)D levels had decreased from 16.6 ng/mL measured 2.3 months before ICU admission to 11 ng/mL thereafter (p<0.01). The author should discuss what is 5.6ngml decrease meaning? Does it mean vitamin D deficiency getting worse in 2.3 months? Or sickening makes total 25OHD lower?
  3. Add 1-2 sentences about low total 25OHD level and vitamin D deficiency in discussion may help for your conclusion.

Author Response

REVIEWER 1

Comments and Suggestions for Authors

The authors addressed most of my comments and the revised manuscript has improved significantly. I have only a few minor comments.

  1. Calcium, ionized calcium, phos and iPTH levels in table 1. are welcome, but do these levels support vitamin D deficiency? The relative low calcium and phos levels with relatively high iPTH levels do suggest true vitamin D deficiency (Jessil et al).

Authors’ reply: We thank the reviewer for this comment. Yes, we agree with the reviewer that the laboratory constellation for all patients with a median 25(OH) D level of 11.6ng/mL (7.2-21.9), decreased total calcium levels (2.01mmol/L [1.87-2.10]), low-normal ionized calcium levels (1.1mmol/L [1.04-1.65]) alongside high-normal iPTH (50.6pg/mL [30.3-115.1]) suggests true Vitamin D deficiency.

However, iPTH levels also appear to have been a little lower than expected given the observed 25(OH) D levels and the median age in our cohort (Ref.: Effects of age and serum 25-OH-vitamin D on serum parathyroid hormone levels.; Valcour A, Blocki F, Hawkins DM, Rao SD; J Clin Endocrinol Metab. 2012 Nov;97(11):3989-95. Epub 2012 Aug 29.). As a possible explanation, critical illness could have aggravated pre-existing VitD deficiency but had not yet resulted in a full counter-regulating iPTH increase.

  1. 25(OH)D levels had decreased from 16.6 ng/mL measured 2.3 months before ICU admission to 11 ng/mL thereafter (p<0.01). The author should discuss what is 5.6ngml decrease meaning? Does it mean vitamin D deficiency getting worse in 2.3 months? Or sickening makes total 25OHD lower?

Authors‘ reply: We thank the reviewer for this question. Supplementing the comment above, the observed decrease in 25(OH) D levels between assessments before and upon ICU admission might have been a consequence of newly onset organ dysfunction (i.e., critical illness). However, we cannot exclude that other factors having acted in the meantime, including dietary intake or sun exposure, contributed to lower 25(OH) D levels following ICU admission. In addition, there was considerable variation in the time between the two measurements in the overall cohort, which renders a robust analysis and subsequent interpretation of results difficult and mostly speculative.

  1. Add 1-2 sentences about low total 25OHD level and vitamin D deficiency in discussion may help for your conclusion.

Authors‘ reply: Concluding the points mentioned above, the following sentences have been added to the discussion section:

Page 8, line 265:

Our observation of lower 25(OH) D levels after ICU admission in patients with pre-admission levels obtained during the previous 12 months could have been attributable to direct interference of critical illness with Vitamin D homeostasis but also to other risk factors (see below) having acted in the meantime. Relatively low calcium and phosphate levels alongside high iPTH levels suggested the presence of true Vitamin D deficiency in our cohort [35]. Still, iPTH levels were slightly (~10%) lower than expected from their known age-dependent relationship with 25(OH) D levels [36]. As one possible explanation integrating these findings, we hypothesize that critical illness might have aggravated pre-existing Vitamin D deficiency in several of our patients.

Finally, we thank the reviewer for the constructive criticism that has allowed us to significantly improve our manuscript. Thank you!

Reviewer 2 Report

To modyfy Table 1 is suggested as the following.

                           25(OH) D

All patients (n=178)

≥20 ng/mL (n=47)

12-20 ng/mL (n=35)

≤12 ng/mL (n=96)

Sex, male

Age, years
